# Interpretable Patterns in Random Initialization Unveil Final Representation

## Abstract

The field of mechanistic interpretability has made strides in unraveling models' hidden representations but is often puzzled by why specific representations form. This paper addresses a crucial question on this front: when a neural network can learn multiple distinct representations to solve a task, how does it "choose" among them during training? We suggest that, at initialization, instead of starting from an empty scratchpad, the model's embedding already contains partially formed representations of varying "completeness." Models tend to develop a representation that is more "complete" at initialization, disregarding less complete alternatives. We empirically examine this hypothesis on algorithmic toy models with clearly defined final representations from which we can elicit an interpretable signal to evaluate such "completeness" of possible representations in the initial embedding. We find that the representations with high initial signals are chosen by the model with high probability, a pattern consistent across models with a single learned representation (remainder equivalence, multi-digit XOR) and with multiple, redundant representations (modular addition). Finally, we investigate the role of embedding dimensionality on model's representation and their "completeness." Our results with toy models show that the seemingly chaotic initialization contains many interpretable patterns to understand the training dynamics of representations.

## 1 Introduction

The field of *mechanistic interpretability* attempts to reverse-engineer the algorithms that neural networks learn. This involves understanding the representations (features) networks learn (Bricken et al., 2023; Cunningham et al., 2023; Gurnee & Tegmark, 2024; Zou et al., 2023) and how these play a role in larger circuits (Elhage et al., 2021; Nanda et al., 2023; Marks et al., 2024). While most such work studies networks as static objects, some have recently begun to study how network representations and circuits form over training (Liu et al., 2022; Olsson et al., 2022; Hoogland et al., 2024; Chen et al., 2023; Singh et al., 2024). By studying training dynamics, we hope to understand not just *what* neural networks learn, but *how* and ultimately *why* they learn the algorithms they do. This understanding may eventually be useful for training models with the properties we desire, such as improved efficiency and safety, through careful design of the training process.

One broad question in mechanistic interpretability regards *universality* (Olah et al., 2020): can models consistently learn the same algorithms across different seeds and scales? While some works have found evidence of universality (Olsson et al., 2022; Gould et al., 2024; Gurnee et al., 2024), in other cases, there seems to be some variability in algorithms and representations that networks learn to solve particular tasks (Zhong et al., 2023; McCoy et al., 2019; Lampinen et al., 2024). In this work, we ask: when networks have a choice between different representations, how do they choose which to learn?

Investigating toy models trained on algorithmic tasks, we can visualize the structure of the final representation forming very early in training using simple tools like PCA, especially under large weight decay. This structure also varies due to different training runs. To understand the reason behind this diversity of representations, we wonder if such structure of the final representation already exists in the seemingly chaotic initialization, albeit not obvious to observe through simple visualization. To formalize this intuition, we propose the following hypothesis:

**Representation "Completeness" Hypothesis** The random embedding initialization already contains many partially formed representations at different levels of "completeness," and the final representation(s) the model learns are those with high "completeness" at initialization. We define "completeness" as an *interpretable* similarity metric that compares such a partially formed representation to its projected, idealized representation, as if it were to be learned by the model.

We empirically evaluate this hypothesis on three algorithmic tasks: remainder equivalence, multi-digit XOR, and modular addition. We analyze these tasks because they have well-defined representations in mathematics. We analyze the first two tasks in §3 as a proof of concept for our hypothesis because the models trained on these tasks are simple and learn a *single* final representation. In §4, we delve deeper into the more interesting case of modular addition, as it is a harder task, and the model learns *multiple, redundant* representations with complicated mechanisms operating on the Fourier space studied extensively by the interpretability community (Liu et al., 2022; Nanda et al., 2023; Zhong et al., 2023; Morwani et al., 2024). We find, in all three cases, we can find an interpretable and intuitive "completeness" metric that projects the structure of the initialization onto the final representation, where high initial "completeness" is a good indicator of this representation being learned by the model.

Lastly, we investigate how embedding dimensionality affects representations in §5. We show that, as the embedding dimension increases, there are fewer representations with low "completeness" in the initialization. In this case, the model can achieve lower test loss when we freeze the embedding at initialization and train only the MLP to fit all the partially complete representations. This helps explain why representation learning could benefit from large embedding dimensions.

Random initialization has been a subject of continuous studies, such as their general downstream capabilities (Amid et al., 2022; Jarrett et al., 2009; Zhong & Andreas, 2024) and the "Lottery Ticket Hypothesis (Frankle & Carbin, 2019)," where sparse subnetworks pruned at initialization achieve similar capabilities. However, our study takes a first step towards understanding how *interpretable patterns for representations* are structured in initialization and how they can evolve to be significant in the final embedding. This understanding is not only driven by scientific curiosity but can also lead to practical benefits, such as speeding up training in our toy algorithmic tasks. We acknowledge that the extension of this approach to non-algorithmic tasks is non-trivial and hence left for future work, but it is encouraging to see that an intuitive understanding of representation dynamics is feasible and such understanding may potentially lead to algorithmic advancement.

## 2 PROBLEM SETUP

### 2.1 TASK SELECTION

Understanding how a model chooses a certain representation is difficult since representations, in general, are high-dimensional and difficult to disentangle, let alone understanding the dynamics between multiple representations. Therefore, we resort to studying toy models, which might provide us with insights into more general models. We specifically need to study models that meet the following criteria: (1) the model has clearly interpretable representation(s) in its embedding, and (2) the model can form non-trivially distinct representations when trained with different initializations. We find three algorithmic tasks that meet such criteria:

**Remainder equivalence** $\mathbb{1}[a = b \mod m]$, where $a, b = 0, 1, \ldots, p - 1$, a number much larger than $m$. The final representation clusters numbers with the same remainder together and forms a line in the embedding, but how the clusters are ordered on the line is up to $p!$ permutations.

**Multi-digit XOR** $a \oplus b = c$. the representation forms a hypercube that each pair of opposite faces have the numbers differ by exactly one bit. However, the exact arrangement of tokens in the hypercube may vary.

**Modular addition** $a + b = c \pmod{p}$, where $a, b, c = 0, 1, \ldots, p - 1$. Prior works have shown circular representations in the embedding, akin to how numbers are arranged around a clock (Power et al., 2022; Liu et al., 2022; 2023; Nanda et al., 2023; Zhong et al., 2023), and that the model learns a few such circles that nicely correspond to Fourier frequencies up to $p$ (Nanda et al., 2023). Over different training runs, different sets of 3-5 frequencies form the final representation, despite all frequencies having a chance to become the final representation.

## 2.2 EXPERIMENT SETUP

We use a fixed model architecture to study all three tasks. Specifically, our model consists of an embedding matrix $\boldsymbol{W}_E$ and a two-layer MLP. The embedding matrix has size $(p, d)$, where $p$ is the vocabulary size and $d$ is the embedding dimension. Every integer $t \in \{0, 1, \ldots, p-1\}$ is treated as a token and has an associated embedding vector $\boldsymbol{E}_t \in \mathbb{R}^d$. For each task, the model tokenizes the two inputs $a$ and $b$, concatenates their embeddings, and feeds $[\boldsymbol{E}_a, \boldsymbol{E}_b] \in \mathbb{R}^{2d}$ to the MLP, which produces a categorical output $c$. Note that we set weight decay to be high in these models to ensure the model quickly forms the final representation.

As our study focuses on the embedding initialization, there are many other factors that influence the representation features, such as initialization of the MLP and dataset training split (Lampinen et al., 2024). Therefore, for our analysis in this paper, we aggregate training results over different random initializations of embeddings, MLPs, and datasets and report the mean and standard deviation.

## 3 "COMPLETENESS" FOR MODELS WITH SINGLE REPRESENTATION

### 3.1 REMAINDER EQUIVALENCE

We train models to learn the task $\mathbb{1}[a = b \mod m]$, given vocabulary size $p = 100$ and fixed divisor $m = 10$. Post-training, as shown in Figure 1(a), token embeddings with the same remainder clustered together, forming a linear arrangement of clusters representing a permutation $\boldsymbol{q}$ of $[0, 1, \ldots, m-1]$.

Observing that the final representation (i.e., the permutation of different remainders) varies across models with different random seeds, we propose a "completeness" metric to capture how well-formed a permutation is at initialization:

$$f(\boldsymbol{q}) = \sum_{0 \leq i < j < m} \left\| \bar{\boldsymbol{E}}_{\boldsymbol{q}_i} - \bar{\boldsymbol{E}}_{\boldsymbol{q}_j} \right\|^2 \cdot (j - i)$$

where

$$\bar{\boldsymbol{E}}_i = \frac{1}{p/m} \sum_{j=0}^{p-1} \mathbf{1}[j \equiv i \;(\mathrm{mod}\; m)] \cdot \boldsymbol{E}_j$$

is the center for each cluster. This metric sums the squared distances between cluster centers, weighted by their relative distances in the permutation.

The metric characterizes permutation "completeness" by considering initial center positions: centers that are close at initialization are likely to remain near each other in the learned representation, and vice versa. The weighted sum is maximized when smaller distances have lower weights and larger distances have higher weights, and we hypothesize that if the initial signal $f(\boldsymbol{q})$ is higher, the model is more likely to learn $\boldsymbol{q}$ as the final representation.

Experimental results support our hypothesis. Figure 1(b) shows that for a given model, the learned representation's initial signal significantly exceeds the average. Across 100 training runs with various random seeds, the learned representation's "completeness" signal consistently ranks within the top 15% on average, with a standard deviation of 22%.

### 3.2 MULTI-DIGIT XOR

For this experiment, the models are trained on the task $c = a \oplus b$, where $a$ and $b$ are 3-bit binary numbers. After training, the embeddings form a cube structure $\alpha$, with each pair of opposite faces differing by exactly one bit. Moreover, different initializations lead to varied token arrangements on the cube. For example, in Figure 2(a), the token 000 shares an edge with token 110 and 001 on the cube in both final embeddings; however, while it also shares an edge with token 011 in Final Embedding 1, the edge is instead shared with token 101 in Final Embedding 2.

As such, we propose another signal metric that measures how close the embedding structure is to a cube of structure $\alpha$:

$$g(\boldsymbol{\alpha}) = \sum_{0 \leq i < j < 2^3} \left\| \boldsymbol{E}_i - \boldsymbol{E}_j \right\|^2 \cdot \boldsymbol{\alpha}_{i,j}$$

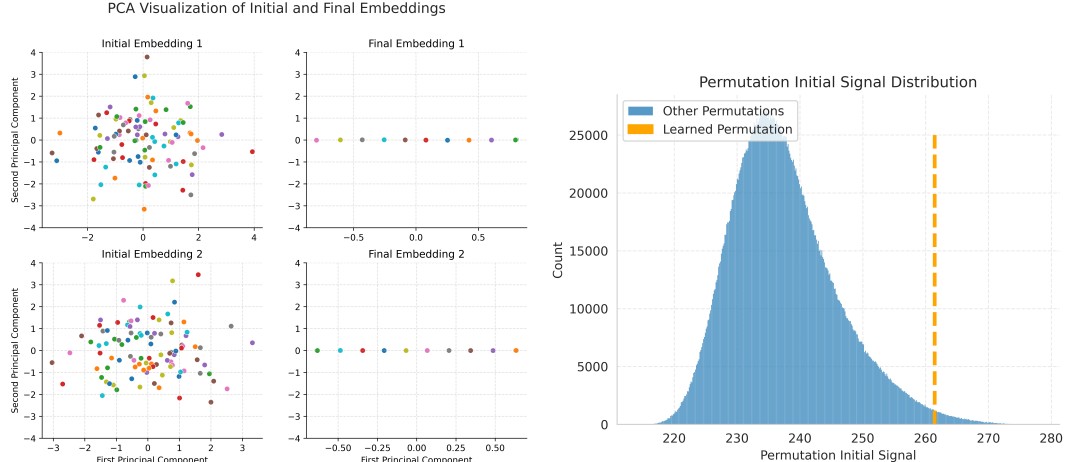

Figure 1: (a) Different colors represent different remainders modulo $m$. The randomly initialized embedding evolves into clusters in a straight line in both models. Notice the order of colors differs across models with different initializations. (b) A histogram of initial "completeness" signals for all the 10! possible permutations, with the signal for the final representation highlighted.

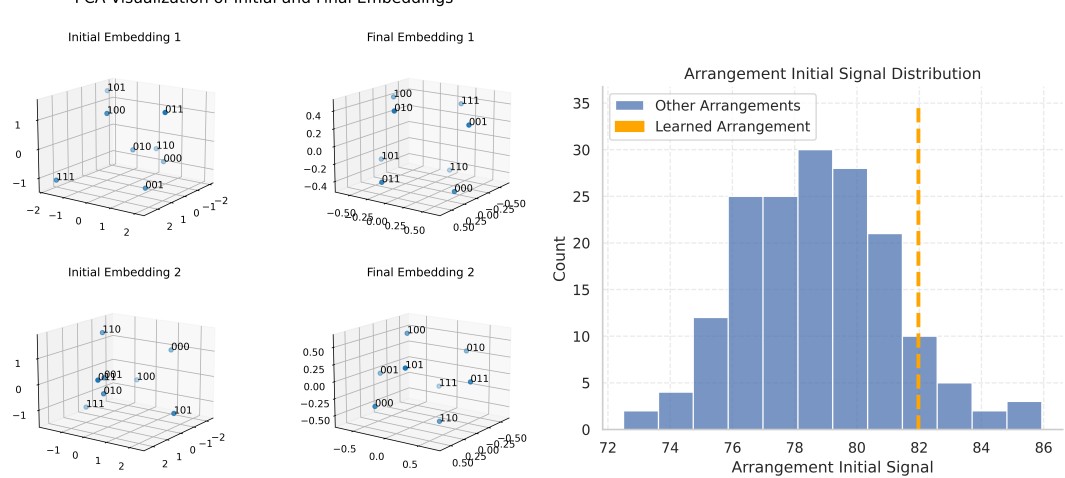

Figure 2: (a) The randomly initialized embedding evolves into cubes in both models. Notice the arrangement of numbers on the cube differs across models with different initializations. (b) Histogram of initial "completeness" signals for all possible 168 arrangements of the cube, with the signal for the final representation highlighted.

Here, $\boldsymbol{\alpha}_{i,j} \in 1, 2, 3$ represents the squared distance between tokens $i$ and $j$ on the unit cube. The function $g(\boldsymbol{\alpha})$ sums the squared distances between token pairs, weighted by their relative cube positions. Borrowing our analysis from §3.1, higher signals $g(\alpha)$ translate to a higher likelihood that $\alpha$ is learned as the final representation.

The result in Figure 2(b) supports this analysis, with learned representations consistently show higher initial signals compared to random arrangements. Across 100 models with different random seeds, the learned representations rank in the top 10% on average, with a standard deviation of 12%.

Both experiments demonstrate that the final learned representations tend to align well with more "complete" structures that we elicit from the initial embedding with high initial signals, supporting our completeness hypothesis. In the following section, we demonstrate the feasibility of our "completeness" hypothesis beyond simple tasks with a single representation. Using modular addition as a

case study, we investigate much more closely whether our hypothesis holds on models with multiple, "redundant" representations as well.

# 4 "COMPLETENESS" FOR MODULAR ADDITION MODEL

## 4.1 REPRESENTATIONS AND THEIR FORMATION

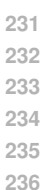

Figure 3: (a) The magnitude of the Fourier coefficients of each embedding frequency over time shown on a logarithmic scale. The surviving frequencies clearly separate themselves from the rest of the frequencies that quickly approach 0. (b) Snapshots of the embedding projected onto a dead frequency (top) and a survived frequency (bottom) at different timesteps during training.

**Circles and Fourier Space**   Prior work has shown that circular representations (circles) are important for neural networks to perform modular addition (Liu et al., 2022; Nanda et al., 2023; Zhong et al., 2023). However, as shown in Figure 4, neighboring numbers along the circle may have increments other than 1 because there are $p$ equivalent group representations corresponding to circles with different Fourier frequencies $k = 1, 2, \cdots, (p-1)/2$. [1] The circle of frequency $k$ places a token $t$ at $(\cos(2\pi kt/p), \sin(2\pi kt/p))$.

Similar to Nanda et al. (2023), we decompose representations into a linear combination of circles of different frequencies. Denoting $\boldsymbol{E}_n \in \mathbb{R}^d$ to be the embedding vector for token $n$, we define the Fourier coefficients of frequency $k$ as

$$\boldsymbol{F}_k = \sum_{n=0}^{p-1} e^{-i2\pi \frac{k}{p} n} \boldsymbol{E}_n. \tag{1}$$

The circle of frequency $k$ is located on the plane spanned by two vectors $\mathrm{Real}(\boldsymbol{E}_k), \mathrm{Imag}(\boldsymbol{E}_k) \in \mathbb{R}^d$.

**Circle Evolution**   We define the Fourier norm of frequency $k$ as $\|\boldsymbol{F}_k\|^2 = \sum_{j=0}^{d-1}(\boldsymbol{F}_k^j)^2$ and observe how the norm of each frequency evolves over the training steps, as shown in Figure 3(a). We observe that only a few circles have significant norm (learned by the model) in the end, while the norm of all other circles decay to almost zero (thus discarded by the model). As shown in Figure 3(c), projections of a surviving frequency stabilize into a clear circle (bottom), while a dead frequency collapses towards its center (top). While prior works have found circles either with Fourier transformation (Nanda et al., 2023) or with PCA (Liu et al., 2022), we use the Fourier transformation because it can better reveal circular representations (see Figure 4). In Figure 3(a), we observe that the surviving circles in the end tend to have higher initial signal. Can we use this information to predict survived circles? We find two "completeness" signals that use information from initialization (and a few training steps out) to predict final representations with high probability: the initial Fourier norm of such frequency and the gradient.

---

[1]Note that frequency $k$ and $p - k$ refer to the same circle, which accounts for the factor 2.

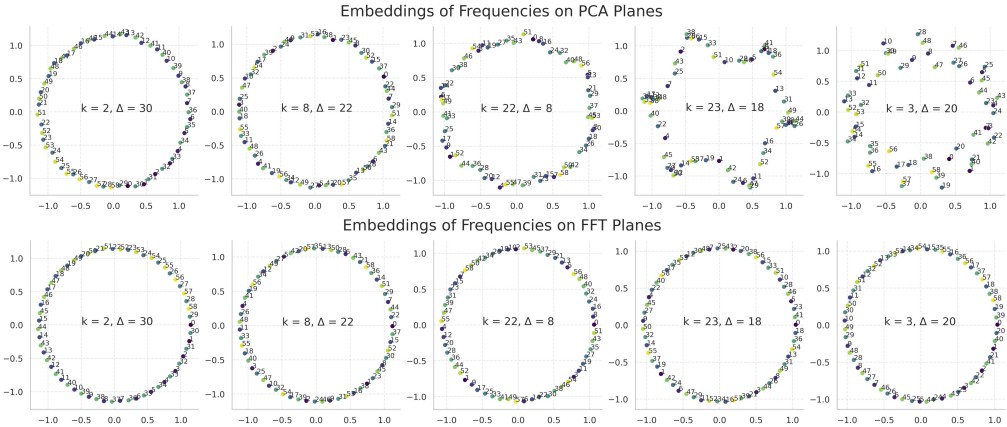

Figure 4: Top: The model embedding projected onto the first 10 principal components in pairs, the only components with significant singular values. Bottom: The model embedding projected onto the Fourier basis of frequencies in descending order of signal magnitude. The $\Delta$ between adjacent tokens shows a correspondence between PCA and FFT.[2] This indicates that PCA provides a loose approximation of circles associated with Fourier frequencies.

## 4.2 INITIAL FOURIER NORM—1ST "COMPLETENESS" SIGNAL

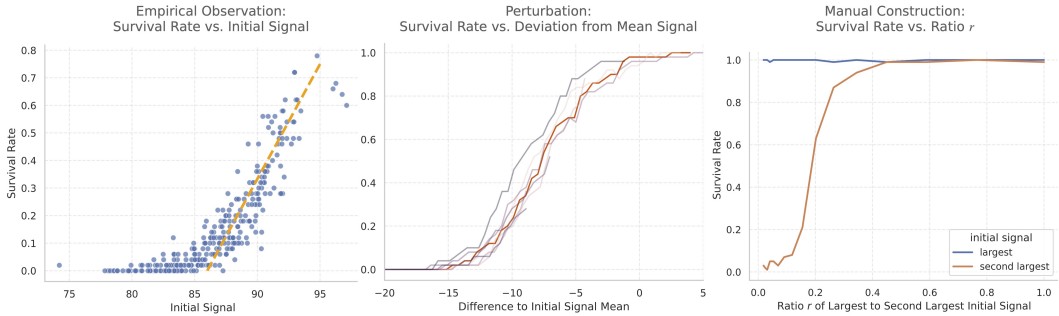

Figure 5: (a) Survival rates of frequencies given their initial Fourier signals over many randomized trials. (b) Survival rate of an arbitrary perturbed frequency versus deviation from the mean initial signal, with different lines showing the same frequency in different embeddings. (c) Survival rates of the largest frequency and the 2nd largest as they differ by $r$.

We define a survival rate as the number of times a particular frequency becomes part of the final representation and hence "survives" over the total number of trials, where we fix the embedding and randomize over different MLPs and dataset initializations. In Figure 5(a), we show that, across 10 different embedding initializations and 50 trials each, there is a linear correlation: the higher the initial signal, the more likely it is for the model to choose that frequency as its final representation. To confirm, we find that the Pearson correlation between survival rate and initial norm is $0.85$, with a p-value smaller than $10^{-3}$.

**Perturbation Experiment**   To corroborate, we conduct a perturbation experiment on the initial embeddings. Specifically, for a given embedding at initialization, we perform a Fourier transform to find the initial coefficients of an arbitrary frequency. We manually enlarge or shrink its magnitude and perform an inverse FFT to restore the embedding, from which we perform model training. We demonstrate in Figure 5(b) the survival rate of the perturbed frequency in different initializations as it deviates from the mean of the rest of the signals. We observe that if the frequency is much higher

---

[2]$\Delta$ can be calculated as the inverse of frequency $k$ modulo $p$ so that $k \cdot \Delta = 1 \ (\mathrm{mod} \ p)$.

than the others, the survival rate is near 100%, while the frequency rarely survives if it is much lower than the mean. This experiment suggests that as we control the environment much more closely, the initial signal of the frequency plays a unique role in determining the final representation.

**Manual Construction**    Moreover, we manually construct an embedding to evaluate the survival rate. We first randomly sample 2 out of $p$ frequencies and set their signals to be of a varying ratio $r \in [0, 1]$. Concretely, the largest frequency will have signal magnitude $s$, while the second highest will have a signal $r \cdot s$. We set all other frequencies to have a signal of small $\epsilon = 10^{-6}$. In this setup, we demonstrate in Figure 5(c) that the frequency with the highest signal always survives, while the second-highest frequency increases in survival rate as its signal increases and differentiates itself further from the other signals. Interestingly, despite all other frequencies having a signal near 0 at initialization, the model sometimes chooses to revive them rather than always choosing the two clear frontrunners, a phenomenon warranting further investigation in the future.

### 4.3   Initial Gradient—2ND "Completeness" Signal

We hypothesize that not only the representations with high initial norms, but also those that can quickly adapts into circles, are more likely to survive. We calculate the gradient as the difference in the norms before and after a given timestep $i$, which takes into account both the embedding gradients and weight decay.

In Figure 6(a), we show the frequencies' initial gradient values alongside whether they "survive" (i.e. become a final representation). Due to the weight decay mechanism, all gradients decrease over time; however, those with higher initial gradients tend to shrink less and are more likely to survive. In Figure 6(b), we show that as the initial gradient increases, the survival rate of these frequencies also increases. To analyze the possibly compounding effect, we show in Figure 6 (c) that frequencies with both high initial norms and gradients are more likely to survive, as the top right corner is more lit with oranges, indicating a greater number of survived frequencies. To further verify this observation, we train a linear support vector machine (SVM) that separates the dead and survived frequencies, achieving an $83.8\%$ accuracy.

### 4.4   Relation to Lottery Ticket Hypothesis

Our analysis can be related to the Lottery Tickets Hypothesis (LTH) (Frankle & Carbin, 2019), where some subnetworks are "winning tickets" that achieve comparable test accuracy to the original network when trained in isolation. Our results suggest that partially formed, decomposable circular representations already exist even at initialization, similar to "winning tickets." Our analysis differs from LTH in two key ways: (1) LTH requires training and pruning to identify "winning" tickets, whereas circles are mathematically defined without training (but specific to modular addition); (2) LTH states only the existence of "winning" tickets at initialization, while we manage to characterize the properties of "winning" circles, that, when trained in isolation, achieve comparable accuracy (see Appendix G). Our work provides representation-level insights into the studies of LTH.

**Faster Training Convergence**    As suggested by Zhang et al. (2021), the pruned "winning ticket" has faster learning convergence, we conduct an ablation experiment in which we keep all frequencies with initial norms one standard deviation above the mean and remove the rest. We train them in separation; given that circles with high initial norms are more likely to be learned, this initialization method significantly accelerates training while still achieving perfect test accuracy. We train the models using 100 random seeds and demonstrate in Figure 6(d) a significant decrease in convergence time, measured by the first time step at which the test accuracy reaches 100%.

## 5   Varying Dimensionality's Role in "Completeness"

After evaluating models with fixed embedding dimension in §3 and §4, we now consider the role of varying dimensionality in the expressive power of model initialization.

Intuitively, a larger model has stronger approximation power and can lead to better performance on the desired task (Sharma & Kaplan, 2020; Michaud et al., 2024; Song et al., 2024). We expect this to hold

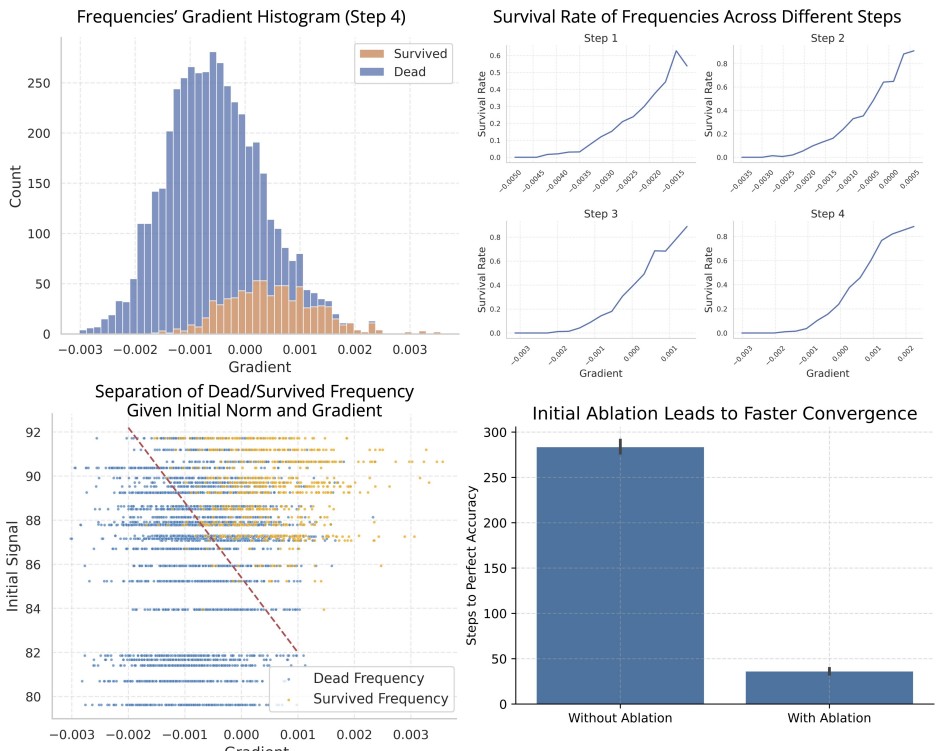

Figure 6: (a) Distribution of the frequency gradient at step 4. (b) Survival rate in relation to frequency gradients for step 1 to 4. (c) Survived (orange) and dead (blue) frequencies characterized by their initial signals and gradients at step 4. (d) Number of steps to reach perfect test accuracy, before and after ablating to only frequencies wit high initial norms, with error bar of 95% confidence interval.

true for our algorithmic models as well. For example, in the context of modular addition, when $d \geq p$, with probability one, perfect circles can be obtained by linearly projecting $d$-dimensional random representations into suitable subspaces. Finding a subspace where a perfect circle of frequency $k$ lives on is equivalent to finding two linear projections $\boldsymbol{p}_1$ and $\boldsymbol{p}_2$ such that $\boldsymbol{E}_a \cdot \boldsymbol{p}_1 = \sin(2\pi ak/p)$ and $\boldsymbol{E}_a \cdot \boldsymbol{p}_2 = \cos(2\pi ak/p)$ for all $a = 0, 1, \cdots, p-1$. Since these equations are linear and linearly independent (due to random initializations), when the number of unknown variables $2d$ is greater (less) than the number of equations $2p$, the system is underdetermined (overdetermined), leading to the existence (nonexistence) of solutions. This conclusion holds for the other two tasks as well under similar reasoning.

**Freeze Embeddings**   To further verify that the initial random embeddings already encapsulate partially complete representations that can solve the problem to some extent, we freeze the embeddings at initialization and train only the MLP. We observe the evolution of test loss over training steps. As dimentionality increases, the test loss decreases at a faster rate during training, as shown in Figure 7 (modular addition) and in Appendix B (remainder equivalence and XOR), confirming that higher-dimensional representations are more "complete" and have a higher approximation power to drive the loss down. More interestingly, regarding the test loss on modular addition in Figure 7, we notice a phase transition around $d = p$—the test loss is significantly lower when $d > p$ compared to when $d < p$.

**Completeness Metrics and Dimensionality**   How does the result of frozen random embeddings solving the algorithmic task relate to the representation "completeness" at initialization? First, it shows that such partially complete representations do exist, and the MLP learns to extract out such patterns, similar to Jarrett et al. (2009). Second, in a standard setting, the MLP would learn an algorithm that computes the task based on the clear embedding representations (line, cube, or circles); however, in the case of the frozen embeddings, the MLP needs to fit an algorithm to all

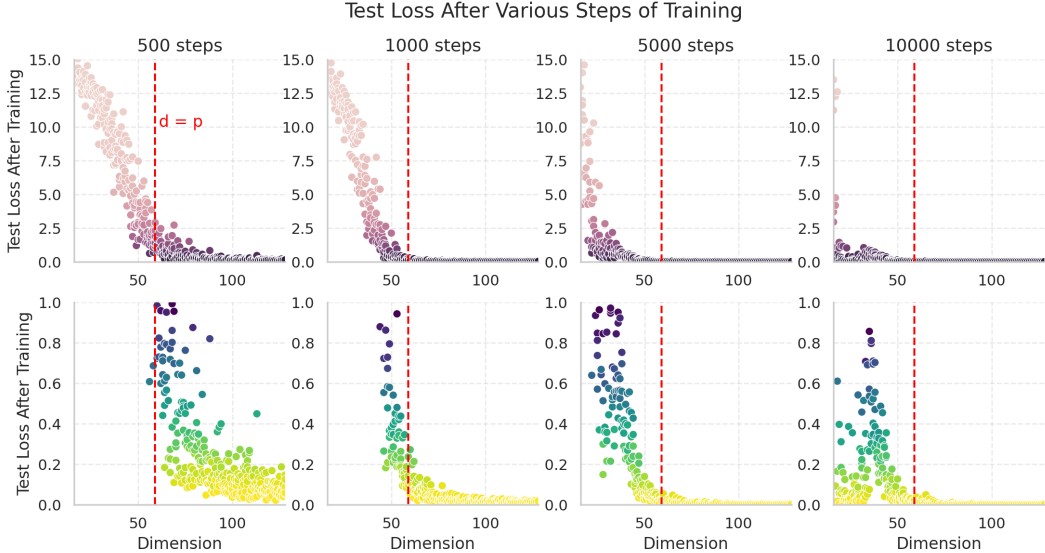

Figure 7: Freezing the initial embedding and training only the MLP, test loss trained on modular addition (zoomed in on the *bottom* to $< 1.0$) in relation to embedding dimension $d$. One can notice a dip in loss at $d = p$. More freeze embedding results in Appendix B.

the partially formed representations there are. We suspect that some particularly poorly formed, "incomplete" representations would serve only as noise and hinder the MLP from fitting the task. We find preliminary evidence to support this by computing the minimum "completeness" score of embeddings with varying dimensions, based on the signal metrics we proposed in §3 and §4. We show in Figure 8 that the minimum "completeness" scores of models on all three tasks increase as the embedding dimension increases, which reduces the noise for fitting the MLP and, thus, increases the test loss shown in Figure 7.

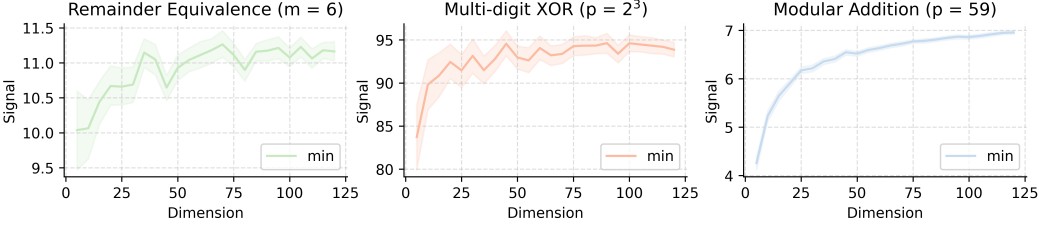

Figure 8: The minimum signal of our constructed "completeness" metrics increases with the dimension.

## 6   RELATED WORK

**Mechanistic Interpretability on Algorithmic Tasks**   Many studies have been done to reverse engineer how neural networks implement algorithmic tasks (Gromov, 2023; Chughtai et al., 2023; Stander et al., 2023; He et al., 2024) because these tasks are mathematically well-defined and simple. However, even on these toy tasks, neural networks display some intriguing phenomena, including phase changes during training (Nanda et al., 2023), different algorithms (Zhong et al., 2023; Liao et al., 2023), or demonstrate the existence of multiple copies of algorithms (Nanda et al., 2023; Zhong et al., 2023). These works on toy models have also extended to similar findings in language models, such as how they compute addition (Zhou et al., 2024; Quirke & Barez, 2024).

**Representation Learning and Variability**    Representation learning is crucial for networks to generalize (Huh et al., 2024; Bengio et al., 2013; Le-Khac et al., 2020; Zou et al., 2023), as paradigms for eliciting better representations get developed (Zhou, 2018; Jaiswal et al., 2020; Grill et al., 2020). Meanwhile, studies have found variability to the learned representations due to random initialization seeds, dataset orders, and representation complexity, leading to downstream consequences (McCoy et al., 2019; Lampinen et al., 2024). On the flipside, prior works have also shown that a single model can contain redundant representations and algorithms (Doimo et al., 2023; Song et al., 2024). Most recent studies have begun to form understandings of representations at a mechanistic level (Li et al., 2023; Engels et al., 2024; Marks & Tegmark, 2023).

**Extracting Patterns from Random Initializations**    Prior work has investigated the capabilities of randomly initialized, untrained networks, particularly convolutional networks, showing that convolutional networks can extract features (Amid et al., 2022; Burda et al., 2018) and achieve comparable downstream performance when linearly combining these random features (Jarrett et al., 2009; Saxe et al., 2011; Cox & Pinto, 2011). More recently, Zhong & Andreas (2024) evaluated the algorithmic capabilities of random transformers, training only the embeddings and fixing everything else. The study shows that random transformers operate in low-dimensional spaces, which corroborates our findings that a "completeness" metric mapping random initialization to a low-dimensional score can explain the model's choice of learned representation.

**Lottery Ticket Hypothesis**    The Lottery Ticket Hypothesis (Frankle & Carbin, 2019) posits that some subnetworks–"winning tickets"–identified at initialization and trained in isolation can match the test accuracy of the original, dense network. It has inspired extensions, such as a stronger conjecture on finding such subnetworks without training (Zhou et al., 2019; Ramanujan et al., 2020; Malach et al., 2020; da Cunha et al., 2022; Orseau et al., 2020; Pensia et al., 2020; Diffenderfer & Kailkhura, 2021), transferring winning tickets across setups (Morcos et al., 2019; Chen et al., 2021), and improving methods of pruning to find the subnetworks (Frankle et al., 2020; Lee et al., 2019; Frankle et al., 2021; Wang et al., 2020; Tanaka et al., 2020). Our work follows a similar style of analysis as the LTH, but our methods of extracting the "winning tickets" are different. While the LTH community focuses on pruning, we propose interpretable (but representation-specific) properties for these partially formed "winning tickets" that allow them to eventually become the learned representations.

# 7    CONCLUSION

In this paper, we propose a Representation "Completeness" Hypothesis that final representations learned by the model often have certain intuitive, interpretable properties at initialization, which make them more "complete" and closer to the ideal representation than other structures not chosen by the model. We verify this hypothesis on three tasks, multi-digit XOR, remainder equivalence, and modular addition, and define such "completeness" metrics on each of the tasks. Finally, we attribute the reason for larger embedding's better ability to generalize to the decrease in ill-formed representations that take training for the model to unlearn, shown by an increase to the lower bound in the "completeness" metrics.

**Limitations**    We have focused on studying algorithmic toy models, as our work is made possible due to their mathematically well-defined representations that can be clearly disentangled. Significant additional work is needed to scale these analyses to even more complex, general models, which remain a challenge.

**Broader Impact**    Understanding training dynamics allows us to better control neural networks in ways we desire, such as making them more accurate and safe. Any dual-use technology, like machine learning models, has accompanying risks, so one should exercise caution when deploying these techniques.

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

## A  TRAINING DETAILS

The MLP in our model consists of two layers: an input layer with dimension $2d$, two hidden layers, each with a width of 100, and an output layer with dimension $p$, which represents the logits for each token from 0 to $p - 1$.

We train the model using the AdamW optimizer (Loshchilov & Hutter, 2019), with a learning rate set to 0.01. The training loss is defined as the cross-entropy loss between the logits computed by the model and the ground truth. To encourage model generalization, we use a train-test split of 80-20 and apply weight decay. The default training details are listed as follows:

| Task | $p$ | $d$ | weight decay | training steps |
|---|---|---|---|---|
| Remainder Equivalence | 100 | 8 | 3 | $10^4$ |
| Multi-digit XOR | 8 | 8 | 3 | $3 \times 10^4$ |
| Modular Addition | 59 | 128 | 0.5 | $3 \times 10^4$ |

In some experiments, we vary $d$, $p$ and weight decay to study their effect on the model.

## B  ADDITIONAL FREEZE EMBEDDING RESULTS

In addition to the modular addition task, we also perform the freeze embedding experiments on the other two tasks and provide the results here. Similar to the phenomenon discussed in §5, the test loss exhibits a decreasing trend with dimensionality.

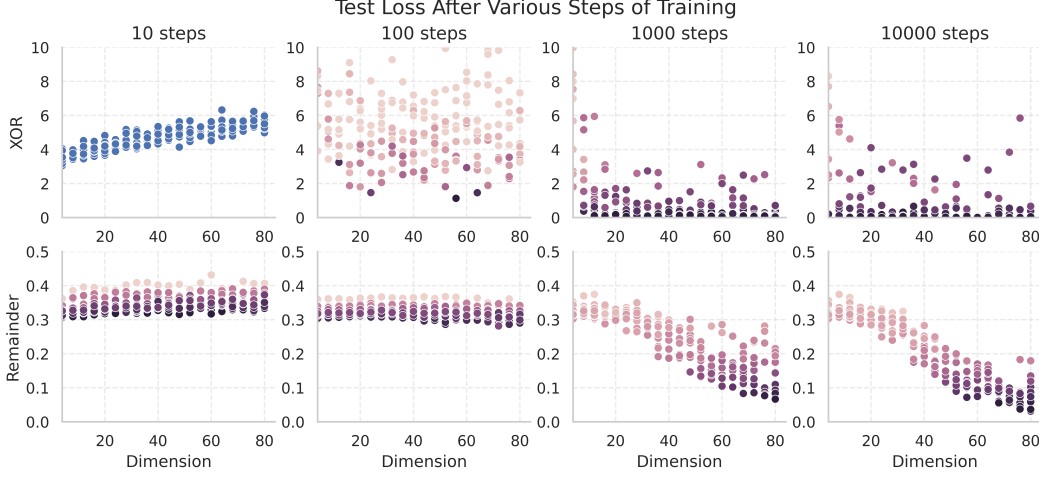

Figure 9: Test loss over time after freezing embedding and training only MLP for XOR (top) and Remainder task (bottom).

## C  DISTRIBUTION OF SIGNAL OVER DIMENSION

Building upon what we analyzed in §5, we compute the distribution of initial "completeness" signals over dimension, sampling from 100 random initializations.

As shown in Figure 10, although the minimum of signals increases, the maximum signal decreases over dimensionality. This indicates that as dimensions increase, the *most complete* representation at initialization is less advantageous, and therefore more representations might have a chance to be learned. This analysis matches our observations in the modular addition task that the number of circular representations increases with dimension, which we elaborate on in Appendix D.

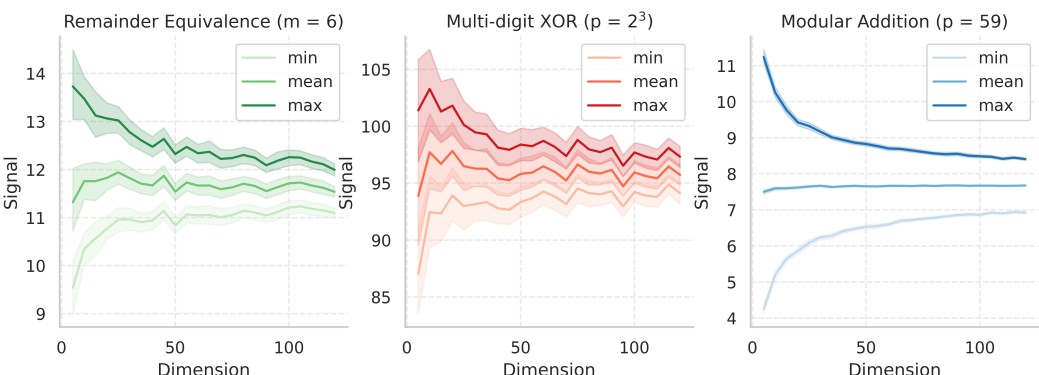

Figure 10: The minimum, average, and maximum of all "completeness" signals over dimension, aggregating over 100 random initializations. The maximum decreases, minimum increases, while the average stays the same.

## D   MULTIPLE REPRESENTATIONS: NUMBER OF CIRCLES

In the main paper, we discuss that the modular addition task exhibits multiple representations, i.e., the model learns multiple circles in the final representation to solve the task. Since dimensionality has a significant impact on the structure of the initial embedding, we want to understand if this factor also impact the number of circles learned. In addition, we study other factors influencing the training dynamics of the toy model, including the modulus $p$ and weight decay.

More specifically, to study the impact of dimensionality $d$ and modulus $p$, we fix one and vary the other to study their effect on the total number of learned circles. Sampling over 100 trials, we identify a clear positive correlation between the size of the embedding $(d, p)$ and number of circles the model learns, as shown in Figure 11[3]. To conclude, as the size of initial embedding $(p \times d)$ increases, the number of circles (algorithm redundancy) increases. This redundant mechanism potentially makes neural networks more robust, but less parameter-efficient and interpretable. Moreover, the result aligns with our result in Appendix C: as the dimensionality of the embedding increases, more representations would have a chance to be learned. As a result, more circles will be learned in the final representation.

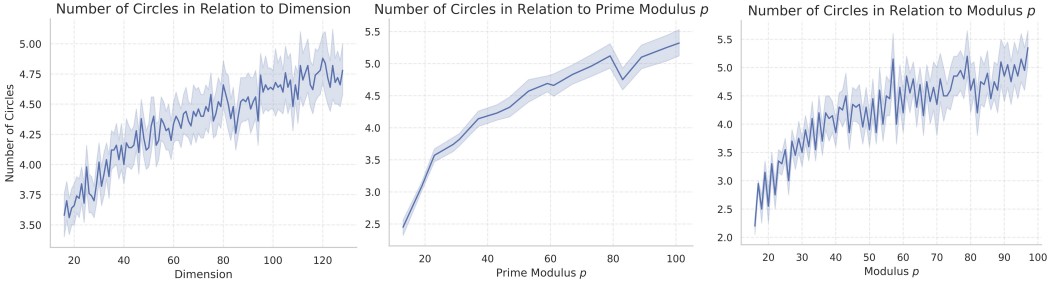

Figure 11: (a): The number of circles as the embedding dimension for representing each token increases from 16 to 128. (b): The number of circles the model chooses for its final representation in relation to the number of tokens, $p$, over 100 random trials, with only prime modulus included. (c): The same figure as (b), but with all modulus $p$ ranging from 16 to 102.

---

[3]Figure 11(b) focuses solely on prime modulus $p$ since non-prime $p$ behaves differently in the modular addition task due to their non-trivial factors. For example, when $p = 12$, a circle with delta $\Delta = 2$ does not cover all the numbers in $[0, p-1]$, and our previous analysis in Fourier basis no longer holds true.

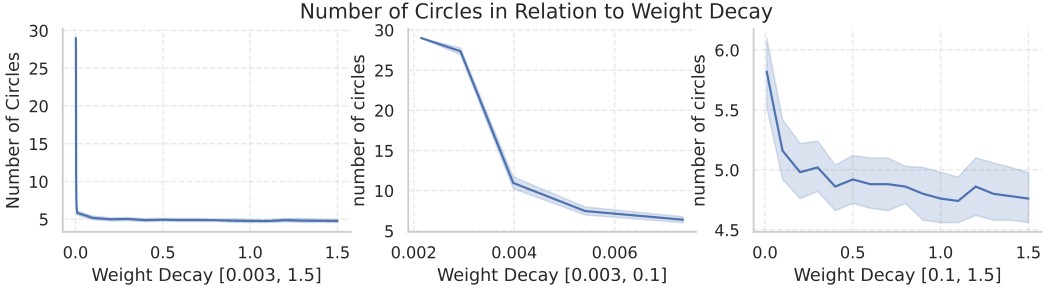

Figure 12: Number of learned circles as a function of weight decay. The *left* panel displays the complete range of weight decays tested in our experiments; the *middle* focuses on smaller weight decays, while the *right* illustrates the transition in the number of learned circles at larger weight decays.

On the other hand, we investigate the impact of weight decay while fixing $d$ and $p$. Considering the extreme case where weight decay is zero, it is not surprising that all the frequencies survive while the neural network fails to generalize, as it has trouble "grokking" (Liu et al., 2022; 2023). As weight decay slightly increases, the number of final circular representations gradually drops, as illustrated in Figure 12.

# E    ALTERNATIVE METRIC FOR MODULAR ADDITION: CIRCULARITY

In addition to the Fourier norm metric used in our discussion of the modular addition task, we compute another metric, circularity, to analyze the initial embedding. We modify the metric introduced in Zhong et al. (2023) by calculating through each frequency in the Fourier Basis instead of through the principal components. As established in the paper, if $X_0, ..., X_{p-1} \in \mathbb{R}^d$ are embeddings projected onto the Fourier basis, the circularity for a specific frequency $f$ is defined as

$$c_k = \frac{2}{p \sum_{j=0}^{p-1} X_{k,j}^2} \left| \sum_{j=0}^{p-1} X_{k,j} e^{2\pi \mathbf{i} \cdot jk/p} \right|^2 \tag{2}$$

Using this metric, we conduct two experiments: one to measure the circularity of the embedding as its dimensionality varies and another to see if initial circularity plays a role in informing eventual representations.

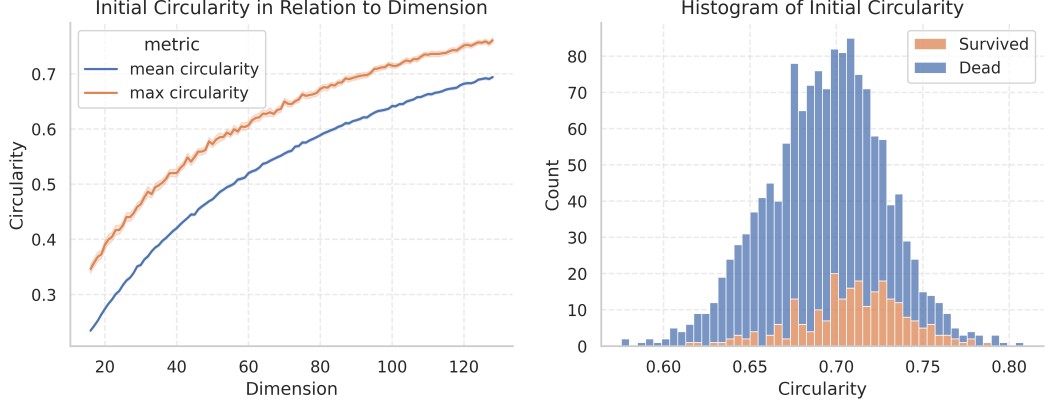

Figure 13: (a) Circularity of projections of the embedding onto different Fourier frequencies at initialization as dimension $d$ varies, calculated using Equation 2. (b) Histogram of different frequencies by their gradients, along with whether they are learned by the model.

To investigate the impact $d$ exerts on the system, we randomly sample 50 initial embeddings of different dimensions and compute the mean and maximum circularity among all frequencies, as shown in Figure 13(a). Indeed, circularity increases as dimensionality increases, confirming our observation that higher-dimensional embedding encodes more complex information at initialization.

We also aim to verify the hypothesis that if the embedding is initialized closer to a circle on a given frequency, that frequency is more likely to survive. However, the evidence shown in Figure 13(b) is inconclusive as to whether larger initial circularity implies a better chance of the representation being learned.

## F    COMPLEX TRAINING DYNAMICS

In our main paper, we propose a metric on the initial embedding to unveil the final model representation. However, we acknowledge that for a specific model, the training dynamics are complicated and dependent on factors other than initialization, including initialization of the MLP, specific train set split, learning rate, weight decay, the hardware the model is trained on, and many more.

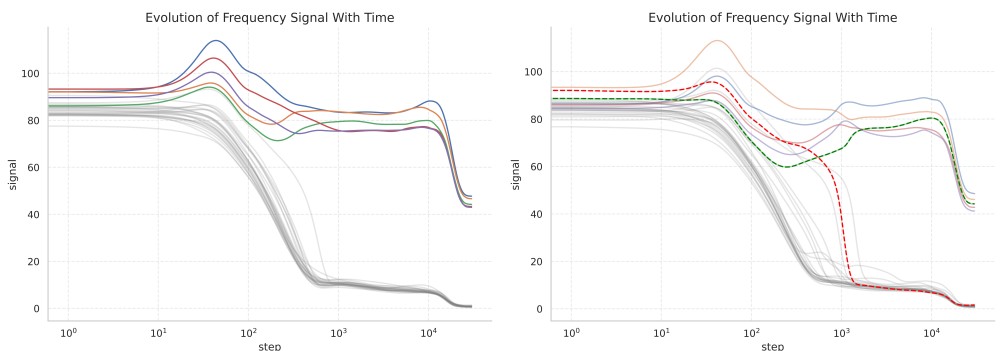

Figure 14: (a): In the modular addition task, the Fourier norm of each embedding frequency evolves over times, as shown in §4. (b): Reproduced (a) on a *different* model, with a high initial norm, dead frequency (dotted red) and a low initial norm frequency that eventually gets learned (dotted green).

As illustrated in Figure 14 (b), a large initial signal does not guarantee that the representation will be learned. Therefore, to mitigate the randomness, we run all the experiments on multiple different random seeds and aggregate the results.

## G    TRAINING ONLY "WINING TICKETS"

Inspired by the Lottery Ticket hypothesis, we are curious if the "Winning Circles" can also reach comparable accuracy when trained separately. Specifically, for a given embedding, we first train without any ablation, from which we identify the original circles the model chooses to learn. We then use $k_1$ to denote the frequency with the largest signal after training and $k_2$ to denote the second-largest frequency.

Using this information, we conduct four experiments to compare the difference between training "Winning Circles" and training random control circles:

A) At initialization, project the embedding onto the Fourier basis, set all frequencies except for $k_1$ to 0, and use inverse FFT to reconstruct the embedding. Train the model using this initial embedding.

B) At initialization, use the same ablation procedure as above, but suppress all other frequencies except for $k_1$ and $k_2$ to 0.

C) At initialization, randomly select a frequency $k_{r,1}$ and suppress all other frequencies except for $k_{r,1}$.

D) At initialization, randomly select two frequencies $k_{r,1}$ and $k_{r,2}$ and suppress all other frequencies except for those two.

In Experiment A, although the embedding is initialized to have only the "Winning Circle", the model revives other circles with an originally $0$ signal and eventually ends up with four learned circles. Figure 15(Left) shows the test loss for Experiment A in *orange*. We can infer from the loss curve that despite trying to learn with only one circle, the model struggles to achieve lower loss with this simple representation and has to make its representation more complex over time, leading to the periodic spikes in test loss. In none of our experiments were we able to construct a model that naturally forms a single-circle representation.

However, in Experiment B, the model achieves good performance using only the two circles initialized with non-zero signals. Two "Winning circles" seem sufficient for the model to achieve a loss as low as $1e^{-7}$. Moreover, Experiment B reaches lower test loss more quickly than the original, mainly because ablating to have only the winning tickets already serves as a type of training and makes the model training process easier, similar to the findings of Zhou et al. (2019).

In comparison, Figure 15(b) shows that the model struggles to learn with only 1 or 2 random circles; Summarizing these results, we conclude that the choice of the circles are not arbitrary: the model can learn the task well with the two "Winning Circles" but not two random circles.

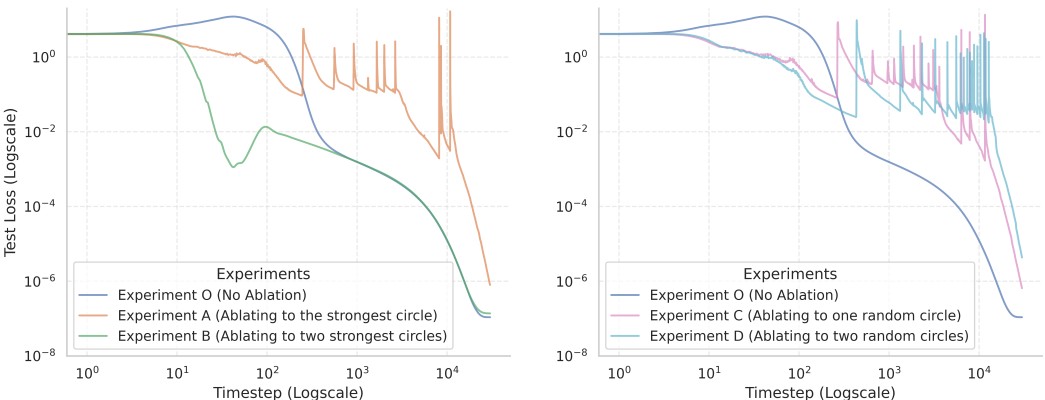

Figure 15: (a) Test loss curve over training timesteps of the ablation experiment when keeping all frequencies (blue), the largest frequency (orange), and the largest two frequencies (green). (b) Test loss curve of the ablation experiment when keeping all frequencies (blue), one random frequency (pink), and two random frequencies (cyan).

Note that in experiments A, C, and D, the test loss curve exhibits several cycles of spiking and subsequent decay. This slingshot phenomenon is associated with the use of the Adam optimizer and often co-occurs with grokking (Thilak et al., 2022).

## H  CHOOSING EMBEDDING GRADIENTS

In Section 4.3, we approximate the gradient for modular addition model as the difference of signals before and after a given timestep. We provide further justification here.

The actual gradient on the embeddings consists of two parts: the gradient $\frac{\partial \mathcal{L}}{\partial E_k}$ produced by MLP on the embedding through backpropagation and weight decay of the initial data. To understand the effect of the former on frequency signals, we use the same Fourier transform procedure to transform the gradients to the Fourier basis and compute their norm, as the Fourier transform is a linear transformation. In Figure 16, we visualize the norm of the gradient in Fourier basis over time. The Fourier gradient spikes around 100 steps and quickly diminishes to near zero after 1000 steps. After this point, the gradient becomes negligibly small, and weight decay becomes the dominant factor affecting the evolution of signals.

These observations motivate experimental decisions in our paper. Specifically, because it is difficult to reconstruct the effect of both backpropagation and weight decay compounded on top of each other on the Fourier basis, especially when weight decay dominates the gradient, we think the difference in signal is a simple and sufficient proxy to conduct experiments with in Section 4.3.

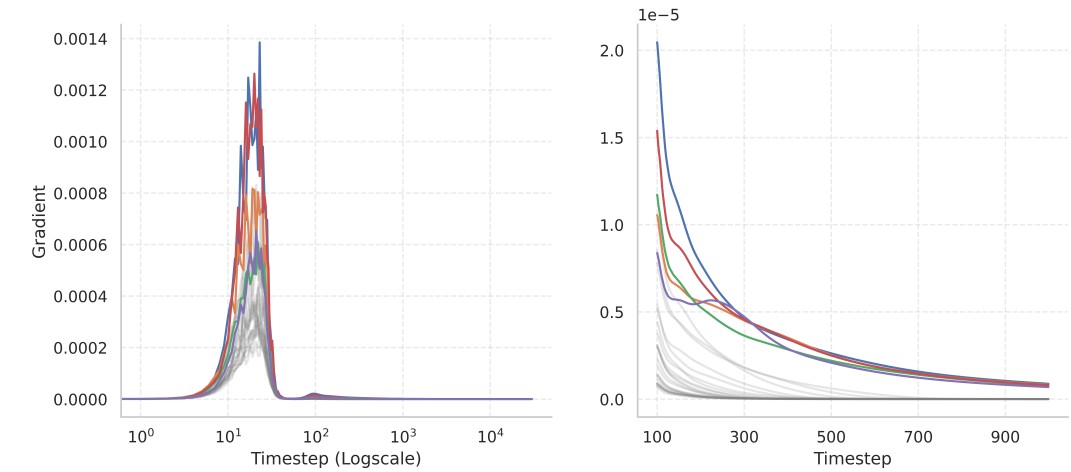

Figure 16: The embedding gradients through backpropagation projected onto the Fourier basis over time. The *right* zooms into timesteps 100 to 1000. The colored curves denote frequencies that eventually are learned, while the grey ones represent gradients of dead frequencies.

# I EXPERIMENTS COMPUTE RESOURCE

All the model training is performed on NVIDIA V100 GPUs. We provide the GPU specs as follows:

- Processor: Intel Xeon Gold 6248
- Nodes: 224
- Clock Rate: 2.5GHz
- CPU cores: 40
- Node RAM: 384GB
- RAM per core: 9GB
- Accelerator type: Nvidia Volta V100
- Accelerators(per Node): 2
- Accelerator RAM: 32GB

The GPU days needed for all the experiments add up to about 68 GPU days. The full research project does not require more compute than the experiments reported in the paper.

