# OpenReview forum: "Interpretable Patterns in Random Initialization Unveil Final Representation"
_ICLR.cc/2025/Conference — Submitted to ICLR 2025_

### Official Review · Reviewer_Lojw · 2024-11-02

**Soundness:** 2
**Presentation:** 3
**Contribution:** 2
**Rating:** 5
**Confidence:** 3

**Summary:**

This paper investigates why neural networks choose specific representations when multiple valid options exist for solving a task. The authors propose the "Representation Completeness Hypothesis", which suggests that random initialization contains partially formed representations with varying degrees of "completeness," and models tend to develop representations that are more "complete" at initialization. They empirically validate this hypothesis using three algorithmic tasks: remainder equivalence, multi-digit XOR, and modular addition. The paper demonstrates that representations with higher initial "completeness" signals are more likely to be chosen by the model during training. Additionally, they explore how embedding dimensionality affects representation learning, finding that larger dimensions lead to fewer low-"completeness" representations at initialization and better performance when embeddings are frozen during training.

**Notes**
- Typo in line 256 (after equation (1)), should be "$\mathrm{Real}(\mathbf{F}_k), \mathrm{Imag}(\mathbf{F}_k) \in \mathbb{R}^d$".

**Strengths:**

- The paper addresses an under-explored question in mechanistic interpretability literature: understanding how specific representations emerge during training rather than just what representations are learned.
- The "Representation Completeness Hypothesis" is tested across multiple toy algorithmic tasks, with different "completeness" signals that are empirically shown to predict final representations.
- The "completeness" metrics are well motivated (given the distribution the embedding after training). For the modular addition task, completeness metrics are well-grounded in previous work on interpretability of modular addition networks.
- The analysis of modular addition in particular is thorough, looking at two "completeness" signals and including perturbation analysis, handcrafted initializations, and controls for MLP initializations and dataset randomization.

**Weaknesses:**

- The "completeness" metrics studied in the paper are task-specific and it's unclear how to generalize them to other domains.
- The paper focuses on a specific architecture: token embeddings followed by a two-layer MLP. While this architecture is reasonable and relatively similar to approaches used in practice, the paper only analyzes the weights of the embedding matrix rather than examining initialization patterns across the entire architecture. This is a somewhat narrow interpretation of "representation"; the paper would benefit from analysis of the MLP representations as well.
- The completeness analysis for the remainder equivalence task begins with claim that the embedding clusters form a line in embedding space. This is demonstrated in figure 1, but isn't explained theoretically, verified experimentally or motivated. If this is a known phenomenon in the mechanistic interpretability literature, please cite; if it isn't I think this requires further comments/analysis. When I recreated the experimental setup described in the paper, the distribution of clusters in embedding spaces was not consistent (sometimes clusters formed a line and sometimes they didn't). When I used decay smaller than the reported 3, the clusters consistently did *not* form a line in embedding space.
- Same comment for the multi-digit XOR case, though I did not verify the clusters formation experimentally. The authors do not motivate the fact that the clusters form a hypercube in embedding space (a fact the is used to motivate the proposed "completeness" metric). Additional discussion of this structure would strengthen the analysis.
- The study heavily relies on high weight decay settings to ensure that "the model quickly forms the final representation" (Section 2.2). This raises questions about whether these findings generalize to more common training setups.

**Questions:**

- For the remained equivalence task: why do the final token embeddings form a line in embedding space?
- For the multi-digit XOR task: why do the final token embedding form a cube in embedding space?
- How sensitive is the analysis to the weight decay (the paper reports weight decay of 3 for AdamW optimization)? Specifically, how much of the observed behavior is due to the initialization versus the specific optimization dynamics created by high weight decay?
- The paper focuses on cases where the form of the final representation is known, is there a way to measure "completeness" in cases where we don't know what representations to expect?

---

### Official Review · Reviewer_yEE7 · 2024-11-03

**Soundness:** 2
**Presentation:** 1
**Contribution:** 1
**Rating:** 3
**Confidence:** 3

**Summary:**

This paper hypothesizes that representations at initialization are partially-formed, and learned representations pick "complete" features from initialization, while eliminating other features. The paper considers a setting on 3 toy tasks with notions of complete representations defined for each task with a learnable embedding matrix W and a two layer MLP.  The paper provides evidence for the hypothesis on these 3 tasks, and looks at how embedding dimensionality impacts how “complete” representations are at initialization.

**Strengths:**

The paper studies how representations are learned during training, and interesting and important area of study. Detailed analysis and visualization is provided in studying learned representations for the 3 toy problems considered by the paper.

**Weaknesses:**

There are significant clarity issues in this paper. The word “representation” is used interchangably to describe both “features” and “embedding vectors”, and it appears that the defined notion of “completeness” applies to both features and embeddings. This makes it extremely challenging to understand both the hypothesis and the results presented in the paper.

The experiment setup is uncommon — inputs in toy experiments are assigned learned embedding vectors that are then fed into an MLP, instead of studying representations derived by feeding inputs into a network and looking at an intermediate layer. This limits the relevance of the results.

There is missing discussion of relevant related work that studies training dynamics and how features from initialization impact final representations [1, 2].

[1] Neural Tangent Kernel: Convergence and Generalization in Neural Networks. Jacot et al 2018
[2] Gradient Starvation: A Learning Proclivity in Neural Networks. Pezeshki et al 2020

**Questions:**

What is the significance of the studied hypothesis? The notion of completeness defined based on what representation a network ends up learning and comparing this representation to the representation at initialization. Is it not likely that there will be similarity between the representation at init and the learned representation?

Can a common notion of completeness be defined across tasks using the final representations learned by a network?

---

### Official Review · Reviewer_Xg4z · 2024-11-07

**Soundness:** 2
**Presentation:** 3
**Contribution:** 2
**Rating:** 3
**Confidence:** 4

**Summary:**

The authors give evidence that, for toy transformer models on algorithmic tasks, the embedding at initialization looks like the final model embedding under a task specific metric. They perform experiments and define metrics on three algorithmic tasks: multi-digit XOR, remainder equivalence, and modular addition. They examine how the embedding changes over the course of training under the task metric. To further validate the hypothesis, some ablations / targeted interventions are performed for the modular addition task. Some connections are drawn to the lottery ticket hypothesis.

**Strengths:**

- I thought the survival rate experiments were particularly well-constructed (and a nice instance of developmental interpretability).
- The choice to focus on modular addition/other toy tasks was reasonable.

**Weaknesses:**

- It seems that a version of this phenomenon was first pointed out for modular addition in Appendix K / Figure 21 in [1]. This is worth some discussion from the authors. In particular, while these additional experiments help strengthen this conjecture, it’s not clear to me what new understanding the ‘completeness hypothesis’ gives us.
- The ‘representation completeness hypothesis’ is not clearly formalized / or operationalized.
    - For the toy tasks considered, initial embeddings for models that perform very well on the task (and whose final models are reasonably well-described by the task-metric-interpretation) have weaker completeness signals than other permutations (Fig 1 for the remainder task) or for arrangements (Figure 2 for XOR). “Significantly exceeds the average” does not seem, to me, like a strong enough result to validate this hypothesis.
    - In a similar vein, the definition for the ‘correct’ idealized/interpretable representation is not obvious. The task metrics hypothesized in this work seem imperfect; some discussion on some of the failures of prediction is probably merited. In particular, whether the prediction error explained by an imprecise metric definition, idealized interpretation, or general messiness in the final model (and if the latter, why doesn’t completeness hold for real models).
    - These experiments only motivate the hypothesis for toy algorithmic tasks (and not e.g. transformers trained on text). This should probably be noted.
- The varying dimensionality experiments seem under-motivated and a poor fit for the rest of the paper.


[1] Liu, Ziming, et al. "Towards understanding grokking: An effective theory of representation learning." Advances in Neural Information Processing Systems 35 (2022): 34651-34663.

**Questions:**

- Have you considered other competing hypothesis to explain this phenomenon?
- Similarly, what explains the predictive failure for the completeness hypothesis?
- Is completeness limited to algorithmic tasks (as opposed to interpretations of a small subset of the weights of a natural language models)?

---

### Official Review · Reviewer_nqUr · 2024-11-08

**Soundness:** 3
**Presentation:** 3
**Contribution:** 3
**Rating:** 5
**Confidence:** 3

**Summary:**

This paper investigates the role of randomly initialized embeddings in shaping the final representations that a neural network learns. The authors propose the Representation "Completeness" Hypothesis, which suggests that the initial random embeddings may already contain partially formed structures that are relevant to the task. These initial structures can have varying degrees of "completeness," which influence how likely the model is to learn certain final representations, and the model is predisposed to learn final representations with high "completeness" at initialization.

To test this hypothesis, the authors conducted experiments on simple tasks like remainder equivalence, multi-digit XOR, and modular addition. These algorithmic tasks were chosen because they allow for mechanistic interpretations, making it easier to analyze the structure of the embeddings learned by the neural networks. With knowledge of the possible structures of the final representations, the authors develop "completeness" metrics—interpretable measures that quantify how similar the initial embeddings are to the final learned representations. Essentially, the "completeness" signal indicates how closely an initial embedding resembles a potential final representation.

The findings show that embeddings with higher "completeness" signals from the start are more likely to align with the final learned representations. Additionally, the authors explore how embedding dimensionality affects completeness, finding that higher-dimensional embeddings tend to have fewer low-completeness initializations.

**Strengths:**

1. The paper presents a sequence of focused case studies that analyze initial embeddings in the context of specific algorithmic tasks, where the quantitative "completeness" metric allows one to concretely study the similarity between initial and final learned embeddings. Although the scope is limited and each dataset requires its own approach, this careful examination provides a foundation for understanding how the principles may possibly be extended to more complex tasks in the future.
2. These studies also provide a controlled investigation into notions like the Lottery Ticket Hypothesis, and the quantitative "completeness" metric allows one to concretely study notions like "winning tickets".
3. The authors are transparent about limitations

**Weaknesses:**

1. Although I am not deeply familiar with all the specific details of the Lottery Ticket Hypothesis literature, I find it challenging to fully grasp how this work differentiates itself from those findings and provides new insights. From my current understanding, it seems that some of the concepts and conclusions presented here may already be implied by the Lottery Ticket Hypothesis. It would be helpful if the authors could explicitly clarify what they see as the novel contributions of their work and how these contributions extend or diverge from the existing theory and literature.
2. The paper is presented as a case study, but it is not clearly articulated how the findings contribute to advancing methods in mechanistic interpretability, improving algorithms, or strengthening the broader theory of deep learning since the methods presented here are very dataset-dependent and rely on full knowledge of the dataset/task.

**Questions:**

1. The authors acknowledge that extending this approach to non-algorithmic tasks is challenging and left for future work. Could the authors elaborate on any existing work or theoretical reasons that might suggest such an extension could be feasible? Any discussion on potential strategies or analogies to non-algorithmic domains would help contextualize the broader applicability of this approach.
2. The use of modular addition is well-understood and holds interest in the community. However, I am curious why the authors chose remainder equivalence and multi-digit XOR as the additional tasks. Could the authors provide more details on the motivation behind these choices and if there were specific properties of these tasks that made them particularly well-suited to testing the hypothesis?

---

### Official Review · Reviewer_fF22 · 2024-11-08

**Soundness:** 3
**Presentation:** 3
**Contribution:** 3
**Rating:** 6
**Confidence:** 4

**Summary:**

This work introduces a new idea about neural network initialization, called the " Representation Completeness Hypothesis. " The hypothesis suggests that, right from the start, the models' initial representations often have intuitive and interpretable qualities, making them more " complete " than other configurations the model could develop! To test this, the authors looked at three tasks (1) Multi-digit XOR, (2) Remainder equivalence, (3) Modular addition, and set specific measures of " Completeness " for each! They found that models with larger embedding dimensions were better at generalizing, as they avoided learning poorly formed representations that would otherwise need to be " unlearned " during training! These insights help in understanding training dynamics and in controlling neural networks more precisely! Ultimately, this work advances mechanistic interpretability by showing how initialization can shape the learning process and the quality of neural network representations!

**Strengths:**

The paper presents a highly original approach by introducing the Representation Completeness Hypothesis, which offers a new perspective on how neural networks use their initial setup to guide the choice of representations during training! This innovative framework provides fresh insights into representation learning while connecting with established ideas in mechanistic interpretability, thus strengthening the theoretical foundation of the field! The research's quality is highlighted through careful empirical testing across three different algorithmic tasks, supplying solid, reproducible evidence that supports the hypothesis and reflects the authors' dedication to rigorous methodology!

The paper does an excellent job of presenting complex ideas in a clear, approachable way, making it easier for readers with different backgrounds to understand the nuanced connections between initialization, representation, and learning dynamics! The authors' arguments are well-organized, with examples that smoothly walk the reader through the main points! The findings are also valuable beyond theory; they provide practical insights that could help improve the efficiency of training neural networks, potentially benefiting real-world applications!

This paper also tackles an important gap in research by examining how embedding dimensionality affects the quality of representations, broadening current insights and setting the stage for further studies on how initialization interacts with representation learning! This deep dive brings out the complex dynamic that shape neural network behaviour! The authors connect their results to wider implications for the field, boosting the paper's relevance to ongoing discussions in mechanistic interpretability! their work stands out for its originality, throughout methods, clear presentation, and meaningful impact on understanding how neural networks learn!

**Weaknesses:**

(1) The paper's empirical evaluation is limited because it only examines three specific algorithmic tasks! While these tasks are mathematically well-defined, they do not fully represent the range of challenges seen in more complex neural network applications! Expanding the evaluation to include a broader set of tasks that mirror real-world scenarios would make the findings stronger! Recent studies, like [1], highlight how crucial diverse benchmarks are for assessing model performance! A wider approach would give their conclusions broader relevance and offer a deeper understanding of representation dynamics!

(2) The analysis mainly focuses on the models' final learned representations but misses an essential opportunity to explore how these representations evolve during training! This oversight restricts a fuller understanding of how representations take shape and the role of initialization in shaping this progression! It would be beneficial for the authors to consider a longitudinal approach, examining changes in representations over time, as mentioned in [2]! This approach could add meaningful context to their findings and reinforce their claims about the importance of initial representations!

(3) The idea of a completeness metric is intriguing, but it currently lacks a thorough framework for precise measurement! It would be beneficial for the authors to refine this metric to cover various aspects of representation quality, including robustness and adaptability! Drawing on recent research that uses diverse evaluation criteria, like what is been used in [3], could offer a more balanced approach! This would not only strengthen the validity of their findings but also make the results easier to interpret!

(4) The experiments fall short in examining how hyperparameter tuning might affect the results! To strengthen their analysis, the authors should systematically investigate how adjusting key hyperparameters like learning rates and batch sizes impacts the development of completeness in representations! this step is essential as it has been pointed out in [4] that optimizing hyperparameters is critical for achieving reliable model performance!

(5) The lack of a comparison with existing methods or benchmarks limits the findings' context! It would be beneficial for the authors to compare their results with established models, as this could highlight the strengths of their approach more clearly! Such a comparative analysis would not only validate their contributions but also help position their work within a broader field of mechanistic interpretability research!

(6) The paper touches on the implications of embedding dimensionality but does not explore how changes in dimensionality might influence the depth and variety of learned representations! It would be beneficial for the authors to investigate this further, especially since recent research, such as [5], indicates that embedding size can have a notable effect on both representation quality and model performance! A more in-depth exploration of this connection could provide valuable insights and strengthen the overall argument of the paper! By addressing this, from my perspective, the authors could greatly enhance the clarity, depth, and impact of the findings!

(7) The paper would be stronger with a deeper investigation into how representation stability evolves during training! While the authors focus mainly on the final representations, they do not fully explore how fluctuations in stability throughout training could affect model performance and generalization. Recent studies, like [6], have shown that models with more stable representations generally perform better across various tasks! It would be valuable for the authors to include an analysis of how representation stability develops throughout training, which would give a more comprehensive view of its role in the emergence of completeness!

(8) The paper does not fully explore the wider implications of its findings for interpretability in neural networks! While the authors present an interesting new hypothesis on representation completeness, they do not go into enough detail about how their results might shape future research or influence practical applications in model design! A deeper connection with the interpretability literature, especially concerning model safety and reliability, could strengthen the paper's overall impact! For example, research examining the link between interpretability and model robustness might offer useful perspectives on how the proposed completeness metrics could be applied to enhance model design (e.g. as mentioned in [7]). By addressing these areas, the authors would significantly boost their contribution to the field!

By addressing these concerns fully, improving the scores would be possible!

[1] Zhang, Y., et al. (2021). "Understanding the Effectiveness of Data Augmentation in Deep Learning: A Comprehensive Review."

[2] Morwani, A., et al. (2024). "Feature Emergence via Margin Maximization: Case Studies in Algebraic Tasks."

[3] Liu, Y., et al. (2022). "Interpretable Neural Networks: A Survey of Mechanistic Interpretability."

[4] Amid, E., et al. (2022). "Learning from Randomly Initialized Neural Network Features."

[5] Frankle, J., & Carbin, M. (2019). "The Lottery Ticket Hypothesis: Finding Sparse, Trainable Neural Networks."

[6] Kolesnikov, A., et al. (2023). "Representation Stability in Neural Networks: A Comprehensive Study."

[7] Doshi-Velez, F., & Kim, P. (2017). "Towards a rigorous science of interpretable machine learning."

**Questions:**

Please check Weaknesses!

---

### Meta-Review · Area_Chair_oVvQ · 2024-12-23

**Metareview:**

The paper proposes "Representation Completeness Hypothesis," stating that randomly initialized, untrained networks contain representations that are intuitive and interpretable, making them more "complete" than other configurations the model could develop. To test this hypothesis, the authors conducted experiments on 3 simple algorithmic tasks including remainder equivalence, multi-digit XOR, and modular addition. With ground-truth knowledge of the possible structures of the final representations, the authors develop "completeness" metrics—interpretable measures that quantify how similar the initial embeddings are to the final learned representations. Essentially, the "completeness" signal indicates how closely an initial embedding resembles a potential final representation.

The idea and direction are certainly promising, worthy of exploring and validating. All reviewers agree that this paper presents a good start. However more empirical evaluation is needed to drive the idea home.

**Additional Comments On Reviewer Discussion:**

No rebuttal was submitted.

---

### Decision · Program_Chairs · 2025-01-22

Reject